# A Flexible Approach to Automated RNN Architecture Generation

**Martin Schrimpf**[*]**, Stephen Merity**[*]**, James Bradbury, & Richard Socher**
Department of Brain and Cognitive Sciences, Massachusetts Institute of Technology
`msch@mit.edu`
Salesforce Research
`{smerity,james.bradbury,rsocher}@salesforce.com`

## Abstract

The process of designing neural architectures requires expert knowledge and extensive trial and error. While automated architecture search may simplify these requirements, the recurrent neural network (RNN) architectures generated by existing methods are limited in both flexibility and components. We propose a domain-specific language (DSL) for use in automated architecture search which can produce novel RNNs of arbitrary depth and width. The DSL is flexible enough to define standard architectures such as the Gated Recurrent Unit and Long Short Term Memory and allows the introduction of non-standard RNN components such as trigonometric curves and layer normalization. Using two different candidate generation techniques, random search with a ranking function and reinforcement learning, we explore the novel architectures produced by the RNN DSL for language modeling and machine translation domains. The resulting architectures do not follow human intuition yet perform well on their targeted tasks, suggesting the space of usable RNN architectures is far larger than previously assumed.

## 1 Introduction

Developing novel neural network architectures is at the core of many recent AI advances (Szegedy et al., 2015; He et al., 2016; Zilly et al., 2016). The process of architecture search and engineering is slow, costly, and laborious. Human experts, guided by intuition, explore an extensive space of potential architectures where even minor modifications can produce unexpected results. Ideally, an automated architecture search algorithm would find the optimal model architecture for a given task.

Many explorations into the automation of machine learning have been made, including the optimization of hyperparameters (Bergstra et al., 2011; Snoek et al., 2012) and various methods of producing novel model architectures (Stanley et al., 2009; Baker et al., 2016; Zoph and Le, 2017). For architecture search, ensuring these automated methods are able to produce results similar to humans usually requires traversing an impractically large search space, assuming high quality architectures exist in the search space at all. The choice of underlying operators composing an architecture is further typically constrained to a standard set across architectures even though recent work has found promising results in the use of non-standard operators (Vaswani et al., 2017).

We propose a meta-learning strategy for flexible automated architecture search of recurrent neural networks (RNNs) which explicitly includes novel operators in the search. It consists of three stages, outlined in Figure 1, for which we instantiate two versions.

1. A candidate architecture generation function produces potential RNN architectures using a highly flexible DSL. The DSL enforces no constraints on the size or complexity of the generated tree and can be incrementally constructed using either a random policy or with an RL agent.
2. A ranking function processes each candidate architecture's DSL via a recursive neural network, predicting the architecture's performance. By unrolling the RNN representation, the ranking function can also model the interactions of a candidate architecture's hidden state over time.

---

[*]Equal contribution. This work was completed while the first author was interning at Salesforce Research.

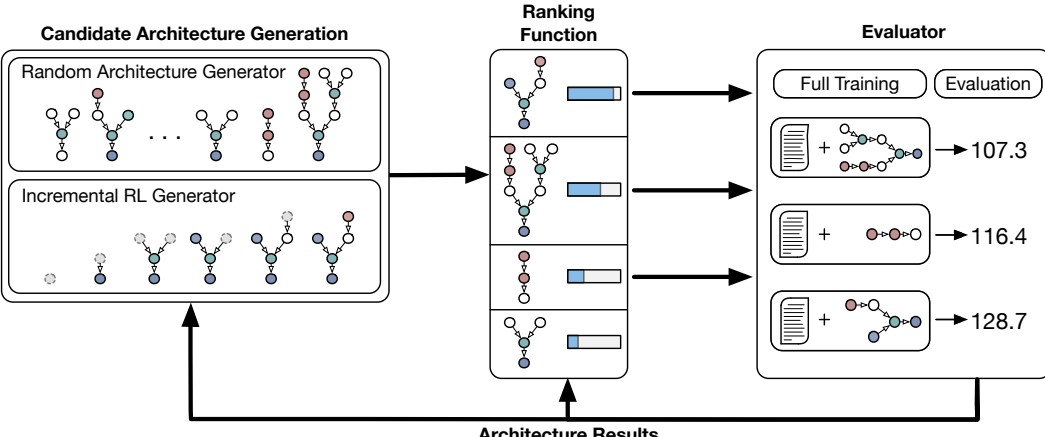

Figure 1: A generator produces candidate architectures by iteratively sampling the next node (either randomly or using an RL agent trained with REINFORCE). Full architectures are processed by a ranking function and the most promising candidates are evaluated. The results from running the model against a baseline experiment are then used to improve the generator and the ranking function.

3. An evaluator, which takes the most promising candidate architectures, compiles their DSLs to executable code and trains each model on a specified task. The results of these evaluations form architecture-performance pairs that are then used to train the ranking function and RL generator.

## 2  A DOMAIN SPECIFIC LANGUAGE FOR DEFINING RNNS

In this section, we describe a domain specific language (DSL) used to define recurrent neural network architectures. This DSL sets out the search space that our candidate generator can traverse during architecture search. In comparison to Zoph and Le (2017), which only produced a binary tree with matrix multiplications at the leaves, our DSL allows a broader modeling search space to be explored.

When defining the search space, we want to allow for standard RNN architectures such as the Gated Recurrent Unit (GRU) (Cho et al., 2014) or Long Short Term Memory (LSTM) (Hochreiter and Schmidhuber, 1997) to be defined in both a human and machine readable manner.

The core operators for the DSL are 4 unary operators, 2 binary operators, and a single ternary operator:

$$[MM, Sigmoid, Tanh, ReLU] \qquad [Add, Mult] \qquad [Gate3].$$

$MM$ represents a single linear layer with bias, i.e. $MM(x) := Wx + b$. Similarly, we define: $Sigmoid(x) = \sigma(x)$. The operator $Mult$ represents element-wise multiplication: $Mult(x, y) = x \circ y$. The $Gate3$ operator performs a weighted summation between two inputs, defined by $Gate3(x, y, f) = \sigma(f) \circ x + (1 - \sigma(f)) \circ y$. These operators are applied to source nodes from the set $[x_t, x_{t-1}, h_{t-1}, c_{t-1}]$, where $x_t$ and $x_{t-1}$ are the input vectors for the current and previous timestep, $h_{t-1}$ is the output of the RNN for the previous timestep, and $c_{t-1}$ is optional long term memory.

The $Gate3$ operator is required as some architectures, such as the GRU, re-use the output of a single $Sigmoid$ for the purposes of gating. While allowing all possible node re-use is out of scope for this DSL, the $Gate3$ ternary operator allows for this frequent use case.

Using this DSL, standard RNN cell architectures such as the $\tanh$ RNN can be defined: $\tanh(Add(MM(x_t), MM(h_{t-1})))$. To illustrate a more complex example that includes $Gate3$, the GRU is defined in full in Appendix A.

### 2.1  ADDING SUPPORT FOR ARCHITECTURES WITH LONG TERM MEMORY

With the operators defined above it is not possible to refer to and re-use an arbitrary node. The best performing RNN architectures however generally use not only a hidden state $h_t$ but also an additional

hidden state $c_t$ for long term memory. The value of $c_t$ is extracted from an internal node computed while producing $h_t$.

The DSL above can be extended to support the use of $c_t$ by numbering the nodes and then specifying which node to extract $c_t$ from (i.e. $c_t = Node_5$). We append the node number to the end of the DSL definition after a delimiter.

As an example, the nodes in bold are used to produce $c_t$, with the number appended at the end indicating the node's number. Nodes are numbered top to bottom ($h_t$ will be largest), left to right.

$$Mult(Sigmoid(MM(x_t)), Tanh(Add(MM(h_{t-1}), \textbf{Mult}(MM(c_{t-1}), MM(x_t)))))|\textbf{5}$$
$$Mult(Sigmoid(MM(x_t)), Tanh(\textbf{Add}(MM(h_{t-1}), Mult(MM(c_{t-1}), MM(x_t)))))|\textbf{6}$$
$$Mult(Sigmoid(MM(x_t)), \textbf{Tanh}(Add(MM(h_{t-1}), Mult(MM(c_{t-1}), MM(x_t)))))|\textbf{7}$$

## 2.2 EXPRESSIBILITY OF THE DOMAIN SPECIFIC LANGUAGE

While the domain specific language is not entirely generic, it is flexible enough to capture most standard RNN architectures. This includes but is not limited to the GRU, LSTM, Minimal Gate Unit (MGU) (Zhou et al., 2016), Quasi-Recurrent Neural Network (QRNN) (Bradbury et al., 2017), Neural Architecture Search Cell (NASCell) (Zoph and Le, 2017), and simple RNNs.

## 2.3 EXTENDING THE DOMAIN SPECIFIC LANGUAGE

While many standard and non-standard RNN architectures can be defined using the core DSL, the promise of automated architecture search is in designing radically novel architectures. Such architectures should be formed not just by removing human bias from the search process but by including operators that have not been sufficiently explored. For our expanded DSL, we include:

$$[Sub, Div] \qquad [Sin, Cos] \qquad [PosEnc] \qquad [LayerNorm, SeLU].$$

These extensions add inverses of currently used operators ($Sub(a, b) = a - b$ instead of addition, $Div(a, b) = \frac{a}{b}$ instead of multiplication), trigonometric curves ($Sin$ and $Cos$ are sine and cosine activations respectively, $PosEnc$ introduces a variable that is the result of applying positional encoding (Vaswani et al., 2017) according to the current timestep), and optimizations ($LayerNorm$ applies layer normalization (Ba et al., 2016) to the input while $SeLU$ is the activation function defined in Klambauer et al. (2017)).

## 2.4 COMPILING A DSL DEFINITION TO EXECUTABLE CODE

For a given architecture definition, we can compile the DSL to code by traversing the tree from the source nodes towards the final node $h_t$. We produce two sets of source code – one for initialization required by a node, such as defining a set of weights for matrix multiplication, and one for the forward call during runtime. For details regarding speed optimizations, refer to Appendix A2.

## 3 CANDIDATE ARCHITECTURE GENERATION

The candidate architecture generator is responsible for producing candidate architectures that are then later filtered and evaluated. Architectures are grown beginning at the output $h_t$ and ordered to prevent multiple representations for equivalent architectures:

**Growing architectures from $h_t$ up**    Beginning from the output node $h_t$, operators are selected to be added to the computation graph, depicted in Figure 2. Whenever an operator has one or more children to be filled, the children are filled in order from left to right. If we wish to place a limit on the height (distance from $h_t$) of the tree, we can force the next child to be one of the source nodes when it would otherwise exceed the maximum height.

**Preventing duplicates through canonical architecture ordering**    Due to the flexibility allowed by the DSL, there exist many DSL specifications that result in the same RNN cell. To solve the

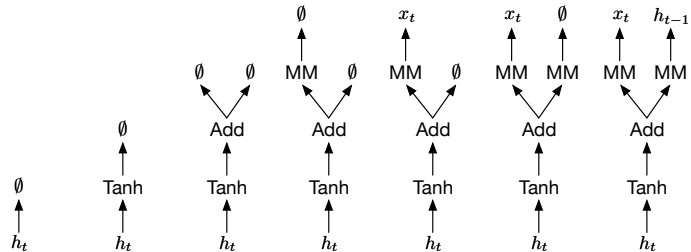

Figure 2: An example of generating an architecture from $h_t$ up. Nodes which have an empty child ($\emptyset$) are filled left to right. A source node such as $x_t$ can be selected at any time if max depth is exceeded.

issue of commutative operators (i.e. $Add(a, b) = Add(b, a)$), we define a canonical ordering of an architecture by sorting the arguments of any commutative nodes. Special consideration is required for non-commutative operators such as *Gate3*, *Sub*, or *Div*. For full details, refer to Appendix A3.

## 3.1 Incremental Architecture Construction using Reinforcement Learning

Architectures in the DSL are constructed incrementally a node at a time starting from the output $h_t$. The simplest agent is a random one which selects the next node from the set of operators without internalizing any knowledge about the architecture or optima in the search space. Allowing an intelligent agent to construct architectures would be preferable as the agent can learn to focus on promising directions in the space of possible architectures.

For an agent to make intelligent decisions regarding which node to select next, it must have a representation of the current state of the architecture and a working memory to direct its actions. We propose achieving this with two components:

1. a tree encoder that represents the current state of the (partial) architecture.
2. an RNN which is fed the current tree state and samples the next node.

The tree encoder is an LSTM applied recursively to a node token and all its children with weights shared, but the state reset between nodes. The RNN is applied on top of the encoded partial architecture and predicts action scores for each operation. We sample with a multinomial and encourage exploration with an epsilon-greedy strategy. Both components of the model are trained jointly using the REINFORCE algorithm (Williams, 1992).

As a partial architecture may contain two or more empty nodes, such as $h_t = Gate3(\emptyset, \emptyset, \sigma(\emptyset))$, we introduce a target token, $T$, which indicates which node is to next be selected. Thus, in $h_t = Gate(T, \emptyset, \sigma(\emptyset))$, the tree encoder understands that the first argument is the slot to be filled.

## 3.2 Filtering Candidate Architectures using a Ranking Function

Even with an intelligent generator, understanding the likely performance of an architecture is difficult, especially the interaction of hidden states such as $h_{t-1}$ and $c_{t-1}$ between timesteps. We propose to approximate the full training of a candidate architecture by training a ranking network through regression on architecture-performance pairs. This ranking function can be specifically constructed to allow a richer representation of the transitions between $c_{t-1}$ and $c_t$.

As the ranking function uses architecture-performance samples as training data, human experts can also inject previous best known architectures into the training dataset. This is not possible for on-policy reinforcement learning and when done using off-policy reinforcement learning additional care and complexity are required for it to be effective (Harutyunyan et al., 2016; Munos et al., 2016).

Given an architecture-performance pair, the ranking function constructs a recursive neural network that reflects the nodes in a candidate RNN architecture one-to-one. Sources nodes are represented by a learned vector and operators are represented by a learned function. The final vector output then passes through a linear activation and attempts to minimize the difference between the predicted and real performance. The source nodes ($x_t$, $x_{t-1}$, $h_{t-1}$, and $c_{t-1}$) are represented by learned vector representations. For the operators in the tree, we use TreeLSTM nodes (Tai et al., 2015). All

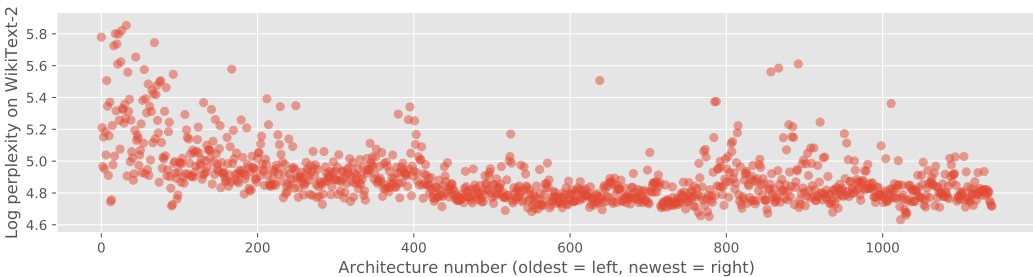

Figure 3: Visualization of the language modeling architecture search over time. Lower log perplexity (y-axis) is better.

operators other than $[Gate3, Sub, Div]$ are commutative and hence can be represented by Child-Sum TreeLSTM nodes. The $[Gate3, Sub, Div]$ operators are represented using an $N$-ary TreeLSTM which allows for ordered children.

**Unrolling the graph for accurately representing $h_{t-1}$ and $c_{t-1}$:**  A strong assumption made above is that the vector representation of the source nodes can accurately represent the contents of the source nodes across a variety of architectures. This may hold true for $x_t$ and $x_{t-1}$ but is not true for $h_{t-1}$ or $c_{t-1}$. The value of $h_t$ and $c_t$ are defined by the operations within the given architecture itself.

To remedy this assumption, we can unroll the architecture for a single timestep, replacing $h_{t-1}$ and $c_{t-1}$ with their relevant graph and subgraph. This would allow the representation of $h_{t-1}$ to understand which source nodes it had access to and which operations were applied to produce $h_{t-1}$.

While unrolling is useful for improving the representation of $h_{t-1}$, it is essential for allowing an accurate representation of $c_{t-1}$. This is as many small variations of $c_{t-1}$ are possible – such as selecting a subgraph before or after an activation – that may result in substantially different architecture performance.

## 4    EXPERIMENTS

We evaluated our architecture generation on two experiments: language modeling (LM) and machine translation (MT). Due to the computational requirements of the experiments, we limited each experiment to one combination of generator components. For language modeling, we explore the core DSL using randomly constructed architectures (random search) directed by a learned ranking function. For machine translation, we use the extended DSL and construct candidate architectures incrementally using the RL generator without a ranking function.

### 4.1    LANGUAGE MODELING USING RANDOM SEARCH WITH A RANKING FUNCTION

For evaluating architectures found during architecture search, we use the WikiText-2 dataset (Merity et al., 2017b). When evaluating a proposed novel RNN cell $c$, we construct a two layer $c$-RNN with a 200 unit hidden size. Aggressive gradient clipping is performed to ensure that architectures such as the ReLU RNN would be able to train without exploding gradients. The weights of the ranking network were trained by regression on architecture-perplexity pairs using the Adam optimizer and mean squared error (MSE). Further hyperparameters and training details are listed in Appendix B1.

**Explicit restrictions on generated architectures**    During the candidate generation phase, we filter the generated architectures based upon specific restrictions. These include structural restrictions and restrictions aimed at effectively reducing the search space by removing likely invalid architectures. For $Gate3$ operations, we force the input to the forget gate to be the result of a sigmoid activation. We also require the cell to use the current timestep $x_t$ and the previous timestep's output $h_{t-1}$ to satisfy the requirements of an RNN. Candidate architectures were limited to 21 nodes, the same number of nodes as used in a GRU, and the maximum allowed distance (height) from $h_t$ was 8 steps.

We also prevent the stacking of two identical operations. While this may be an aggressive filter it successfully removes many problematic architectures. These problematic architectures include when two sigmoid activations, two ReLU activations, or two matrix multiplications are used in succession – the first of which is unlikely to be useful, the second of which is a null operator on the second activation, and the third of which can be mathematically rewritten as a single matrix multiplication.

If a given candidate architecture definition contained $c_{t-1}$, the architecture was queried for valid subgraphs from which $c_t$ could be generated. The subgraphs must contain $c_{t-1}$ such that $c_t$ is recurrent and must contain three or more nodes to prevent trivial recurrent connections. A new candidate architecture is then generated for each valid $c_t$ subgraph.

**Random architecture search directed by a learned ranking function**    Up to 50,000 candidate architecture DSL definitions are produced by a random architecture generator at the beginning of each search step. This full set of candidate architectures are then simulated by the ranking network and an estimated perplexity assigned to each. Given the relative simplicity and small training dataset, the ranking function was retrained on the previous full training results before being used to estimate the next batch of candidate architectures. Up to 32 architectures were then selected for full training. 28 of these were selected from the candidate architectures with the best perplexity while the last 4 were selected via weighted sampling without replacement, prioritizing architectures with better estimated perplexities.

$c_t$ architectures were introduced part way through the architecture search after 750 valid $h_t$ architectures had been evaluated with $h_t$ architectures being used to bootstrap the $c_t$ architecture vector representations. Figure 3 provides a visualization of the architecture search over time, showing valid $h_t$ and $c_t$ architectures.

**Analyzing the BC3 cell**    After evaluating the top 10 cells using a larger model on WikiText-2, the top performing cell BC3 (named after the identifying hash, `bc3dc7a...`) was an unexpected layering of two $Gate3$ operators,

$$f = \sigma(W^f x_t + U^f h_{t-1}) \tag{1}$$
$$z = V^z(X^y c_{t-1} \circ U^z x_t) \circ W^z x_t \tag{2}$$
$$c_t = \tanh(f \circ W^g x_t + (1 - f) \circ z) \tag{3}$$

$$o = \sigma(W^o x_t + U^o h_{t-1}) \tag{4}$$
$$h_t = o \circ c_t + (1 - o) \circ h_{t-1} \tag{5}$$

where $\circ$ is an element-wise multiplication and all weight matrices $W, U, V, X \in \mathbb{R}^{H \times H}$.

Equations 1 to 3 produce the first $Gate3$ while equations 4 and 5 produce the second $Gate3$. The output of the first $Gate3$ becomes the value for $c_t$ after passing through a $\tanh$ activation.

While only the core DSL was used, BC3 still breaks with many human intuitions regarding RNN architectures. While the formulation of the gates $f$ and $o$ are standard in many RNN architectures, the rest of the architecture is less intuitive. The $Gate3$ that produces $c_t$ (equation 3) is mixing between a matrix multiplication of the current input $x_t$ and a complex interaction between $c_{t-1}$ and $x_t$ (equation 2). In BC3, $c_{t-1}$ passes through multiple matrix multiplications, a gate, and a $\tanh$ activation before becoming $c_t$. This is non-conventional as most RNN architectures allow $c_{t-1}$ to become $c_t$ directly, usually through a gating operation. The architecture also does not feature a masking output gate like the LSTM, with outputs more similar to that of the GRU that does poorly on language modeling. That this architecture would be able to learn without severe instability or succumbing to exploding gradients is not intuitively obvious.

### 4.1.1 EVALUATING THE BC3 CELL

For the final results on BC3, we use the experimental setup from Merity et al. (2017a) and report results for the Penn Treebank (Table 1) and WikiText-2 (Table 2) datasets. To show that not any standard RNN can achieve similar perplexity on the given setup, we also implemented and tuned a GRU based model which we found to strongly underperform compared to the LSTM, BC3, NASCell, or Recurrent Highway Network (RHN). Full hyperparameters for the GRU and BC3 are in Appendix

| Model | Parameters | Validation | Test |
|---|---|---|---|
| Inan et al. (2016) - Variational LSTM (tied) + augmented loss | 24M | 75.7 | 73.2 |
| Inan et al. (2016) - Variational LSTM (tied) + augmented loss | 51M | 71.1 | 68.5 |
| Zilly et al. (2016) - Variational RHN (tied) | 23M | 67.9 | 65.4 |
| Zoph and Le (2017) - Variational NAS Cell (tied) | 25M | – | 64.0 |
| Zoph and Le (2017) - Variational NAS Cell (tied) | 54M | – | 62.4 |
| Melis et al. (2017) - 4-layer skip connection LSTM (tied) | 24M | 60.9 | 58.3 |
| Merity et al. (2017a) - 3-layer weight drop LSTM + NT-ASGD (tied) | 24M | 60.0 | 57.3 |
| 3-layer weight drop GRU + NT-ASGD (tied) | 24M | 76.1 | 74.0 |
| 3-layer weight drop BC3 + NT-ASGD (tied) | 24M | 64.4 | 61.4 |

Table 1: Model perplexity on validation/test sets for the Penn Treebank language modeling task.

| Model | Parameters | Validation | Test |
|---|---|---|---|
| Inan et al. (2016) - Variational LSTM (tied) | 28M | 92.3 | 87.7 |
| Inan et al. (2016) - Variational LSTM (tied) + augmented loss | 28M | 91.5 | 87.0 |
| Melis et al. (2017) - 1-layer LSTM (tied) | 24M | 69.3 | 65.9 |
| Melis et al. (2017) - 2-layer skip connection LSTM (tied) | 24M | 69.1 | 65.9 |
| Merity et al. (2017a) - 3-layer weight drop LSTM + NT-ASGD (tied) | 33M | 68.6 | 65.8 |
| 3-layer weight drop GRU + NT-ASGD (tied) | 33M | 92.2 | 89.1 |
| 3-layer weight drop BC3 + NT-ASGD (tied) | 33M | 79.7 | 76.4 |

Table 2: Model perplexity on validation/test sets for the WikiText-2 language modeling task.

B4. Our model uses equal or fewer parameters compared to the models it is compared against. While BC3 did not outperform the highly tuned AWD-LSTM (Merity et al., 2017a) or skip connection LSTM (Melis et al., 2017), it did outperform the Recurrent Highway Network (Zilly et al., 2016) and NASCell (Zoph and Le, 2017) on the Penn Treebank, where NASCell is an RNN found using reinforcement learning architecture search specifically optimized over the Penn Treebank.

## 4.2 Incremental Architecture Construction using RL for Machine Translation

For our experiments involving the extended DSL and our RL based generator, we use machine translation as our domain. The candidate architectures produced by the RL agent were directly used without the assistance of a ranking function. This leads to a different kind of generator: whereas the ranking function learns global knowledge about the whole architecture, the RL agent is trimmed towards local knowledge about which operator is ideal to be next.

**Training details**    Before evaluating the constructed architectures, we pre-train our generator to internalize intuitive priors. These priors include enforcing well formed RNNs (i.e. ensuring $x_t$, $h_{t-1}$, and one or more matrix multiplications and activations are used) and moderate depth restrictions (between 3 and 11 nodes deep). The full list of priors and model details are in Appendix C1.

For the model evaluation, we ran up to 28 architectures in parallel, optimizing one batch after receiving results from at least four architectures. As failing architectures (such as those with exploding gradients) return early, we needed to ensure the batch contained a mix of both positive and negative results. To ensure the generator yielded mostly functioning architectures whilst understanding the negative impact of invalid architectures, we chose to require at least three good architectures with a maximum of one failing architecture per batch.

For candidate architectures with multiple placement options for the memory gate $c_t$, we evaluated all possible locations and waited until we had received the results for all variations. The best $c_t$ architecture result was then used as the reward for the architecture.

**Baseline Machine Translation Experiment Details**    To ensure our baseline experiment was fast enough to evaluate many candidate architectures, we used the Multi30k English to German (Elliott et al., 2016) machine translation dataset. The training set consists of 30,000 sentence pairs that briefly

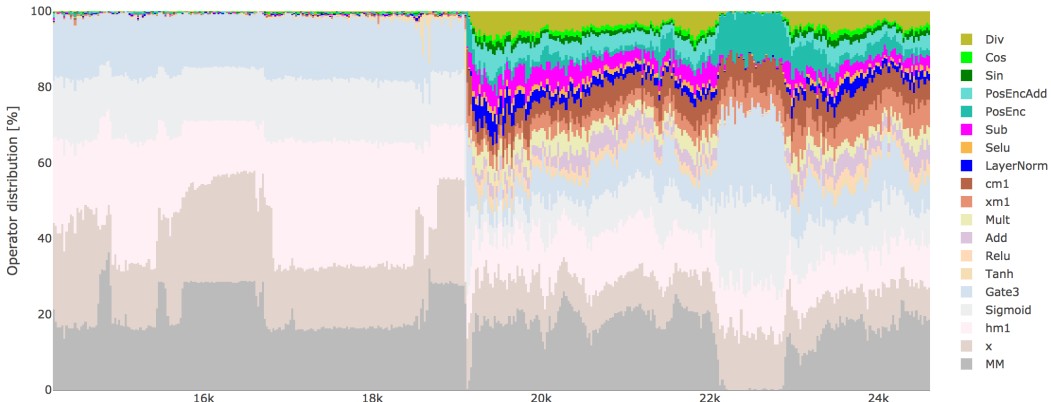

Figure 4: Distribution of operators over time. Initially the generator primarily uses the core DSL (faded colors) but begins using the extended DSL as the architecture representation stabilizes.

describe Flickr captions. Our experiments are based on OpenNMT codebase with an attentional unidirectional encoder-decoder LSTM architecture, where we specifically replace the LSTM encoder with architectures designed using the extend DSL.

For the hyper-parameters in our baseline experiment, we use a hidden and word encoding size of 300, 2 layers for the encoder and decoder RNNs, batch size of 64, back-propagation through time of 35 timesteps, dropout of 0.2, input feeding, and stochastic gradient descent. The learning rate starts at 1 and decays by 50% when validation perplexity fails to improve. Training stops when the learning rate drops below 0.03.

**Analysis of the Machine Translation Architecture Search**    Figure 4 shows the relative frequency of each operator in the architectures that were used to optimize the generator each batch. For all the architectures in a batch, we sum up the absolute number that each operator occurs and divide by the total number of operators in all architectures of the batch. By doing this for all batches (x-axis), we can see which operators the generator prefers over time.

Intriguingly, the generator seems to rely almost exclusively on the core DSL ($MM, Gate3, Tanh, Sigmoid, x_t, h_{t-1}$) when generating early batches.   The low usage of the extended DSL operators may also be due to these operators frequently resulting in unstable architectures, thus being ignored in early batches. Part way through training however the generator begins successfully using a wide variety of the extended DSL ($Sub, Div, Sin, Cos, \ldots$).   We hypothesize that the generator first learns to build robust architectures and is only then capable of inserting more varied operators without compromising the RNN's overall stability. Since the reward function it is fitting is complex and unknown to the generator, it requires substantial training time before the generator can understand how robust architectures are structured. However, the generator seems to view the extended DSL as beneficial given it continues using these operators.

Overall, the generator found 806 architectures that out-performed the LSTM based on raw test BLEU score, out of a total of 3450 evaluated architectures (23%).   The best architecture (determined by the validation BLEU score) achieved a test BLEU score of 36.53 respectively, compared to the standard LSTM's 34.05.   Multiple cells also rediscovered a variant of residual networks ($Add(Transformation(x_t), x_t)$) (He et al., 2016) or highway networks ($Gate3(Transformation(x_t), x_t, Sigmoid(\ldots))$) (Srivastava et al., 2015). Every operation in the core and extended DSL made their way into an architecture that outperformed the LSTM and many of the architectures found by the generator would likely not be considered valid by the standards of current architectures. Figure 5 highlights how often the full range of operators occur in architectures that out-performed the LSTM. These results suggest that the space of successful RNN architectures might hold many unexplored combinations with human bias possibly preventing their discovery.

In Table 3 we take the top five architectures found during automated architecture search on the Multi30k dataset and test them over the IWSLT 2016 (English to German) dataset (Cettolo et al., 2016).  The training set consists of 209,772 sentence pairs from transcribed TED presentations

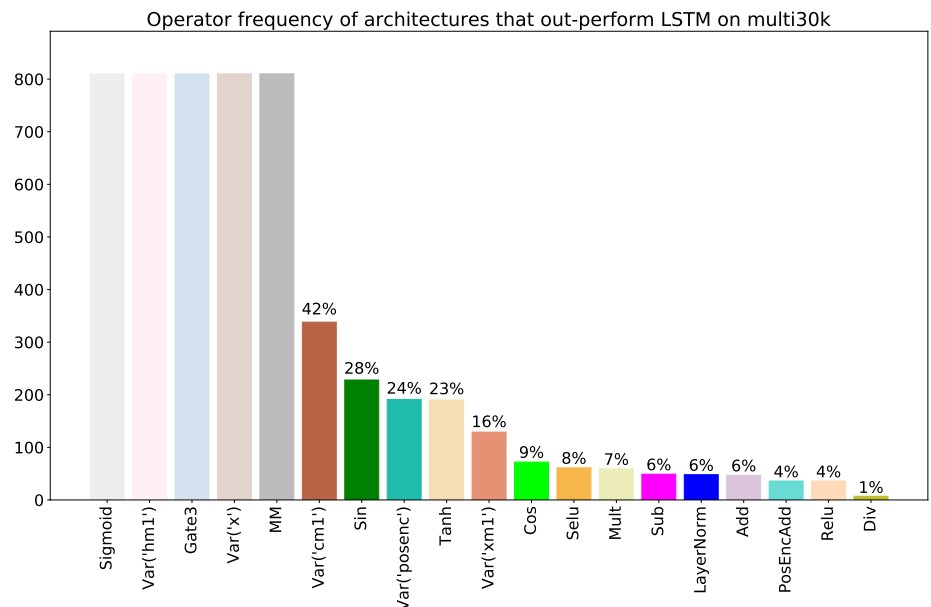

Figure 5: Operator frequency of architectures that out-perform LSTM on Multi30k (colored like Fig. 4). For every architecture with a BLEU score higher than LSTM, we count if an operator occurs in its architecture. While variables $x_t$ and $h_{t-1}$ are inherent to every architecture, the generator also picked up on the $Gate3 - Sigmoid$ combination for every one of its top architectures. Intriguingly, even operators that are less commonly used in the field such as sine curves and positional encoding occur in a large number of architectures and thus seem to contribute to successful architectures.

| DSL Architecture Description | Multi30k | | IWSLT'16 |
| (encoder used in attentional encoder-decoder LSTM) | Val Loss | Test BLEU | Test BLEU |
|---|---|---|---|
| LSTM | 6.05 | 34.05 | 24.39 |
| BC3 | 5.98 | 34.69 | 23.32 |
| $Gate3(Add(MM(x_t), x_t), Sigmoid(MM(h_{t-1})))$ | 5.83 | 36.03 | 22.76 |
| $Gate3(MM(x_t), Tanh(x_t), Sigmoid(MM(h_{t-1})))$ | 5.82 | 36.20 | 22.99 |
| 5-deep nested $Gate3$ with $LayerNorm$ (see Appendix C2) | 5.66 | 36.35 | 22.53 |
| Residual $x_t$ with positional encoding (see Appendix C2) | 5.65 | 35.63 | 22.48 |
| $Gate3(MM(x_t), x_t, Sigmoid(MM(SeLU(h_{t-1}))))$ | 5.64 | 36.53 | 23.04 |

Table 3: Model loss and BLEU on the Multi30k and IWSLT'16 MT datasets. All architectures were generated on the Multi30k dataset other than the LSTM and BC3 from the LM architecture search. We did not perform any hyperparameter optimizations on either dataset to avoid unfair comparisons, though the initial OpenNMT hyperparameters likely favored the baseline LSTM model.

that cover a wide variety of topics with more conversational language than in the Multi30k dataset. This dataset is larger, both in number of sentences and vocabulary, and was not seen during the architecture search. While all six architectures achieved higher validation and test BLEU on Multi30k than the LSTM baseline, it appears the architectures did not transfer cleanly to the larger IWSLT dataset. This suggests that architecture search should be either run on larger datasets to begin with (a computationally expensive proposition) or evaluated over multiple datasets if the aim is to produce general architectures. We also found that the correlation between loss and BLEU is far from optimal: architectures performing exceptionally well on the loss sometimes scored poorly on BLEU. It is also unclear how these metrics generalize to perceived human quality of the model (Tan et al., 2015) and thus using a qualitatively and quantitatively more accurate metric is likely to benefit the generator. For hyper parameters of the IWSLT model, refer to Appendix C3.

## 5 RELATED WORK

Architecture engineering has a long history, with many traditional explorations involving a large amount of computing resources and an extensive exploration of hyperparamters (Jozefowicz et al., 2015; Greff et al., 2016; Britz et al., 2017). The approach most similar to our work is Zoph and Le (2017) which introduces a policy gradient approach to search for convolutional and recurrent neural architectures. Their approach to generating recurrent neural networks was slot filling, where element-wise operations were selected for the nodes of a binary tree of specific size. The node to produce $c_t$ was selected once all slots had been filled. This slot filling approach is not highly flexible in regards to the architectures it allows. As opposed to our DSL, it is not possible to have matrix multiplications on internal nodes, inputs can only be used at the bottom of the tree, and there is no complex representation of the hidden states $h_{t-1}$ or $c_{t-1}$ as our unrolling ranking function provides. Many other similar techniques utilizing reinforcement learning approaches have emerged such as designing CNN architectures with Q-learning (Baker et al., 2016).

Neuroevolution techniques such as NeuroEvolution of Augmenting Topologies (NEAT) (Stanley and Miikkulainen, 2002) and HyperNEAT (Stanley et al., 2009) evolve the weight parameters and structures of neural networks. These techniques have been extended to producing the non-shared weights for an LSTM from a small neural network (Ha et al., 2016) and evolving the structure of a network (Fernando et al., 2016; Bayer et al., 2009).

## 6 CONCLUSION

We introduced a flexible domain specific language for defining recurrent neural network architectures that can represent most human designed architectures. It is this flexibility that allowed our generators to come up with novel combinations in two tasks. These architectures used both core operators that are already used in current architectures as well as operators that are largely unstudied such as division or sine curves. The resulting architectures do not follow human intuition yet perform well on their targeted tasks, suggesting the space of usable RNN architectures is far larger than previously assumed. We also introduce a component-based concept for architecture search from which we instantiated two approaches: a ranking function driven search which allows for richer representations of complex RNN architectures that involve long term memory ($c_t$) nodes, and a Reinforcement Learning agent that internalizes knowledge about the search space to propose increasingly better architectures. As computing resources continue to grow, we see automated architecture generation as a promising avenue for future research.

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

## APPENDIX A: DOMAIN SPECIFIC LANGUAGE

**DSL GRU Definition**

```
Gate3(
    Tanh(
        Add(
            MM($x_t$),
            Mult(
                MM($h_{t-1}$),
                Sigmoid(
                    Add( MM($h_{t-1}$), MM($x_t$) )
                )
            )
        )
    ),
    $h_{t-1}$,
    Sigmoid(
        Add( MM($h_{t-1}$), MM($x_t$) ),
    )
)
```

**DSL BC3 Definition**

```
Gate3(
    Tanh(
        Gate3(
            MM($x_t$),
            Mult(
                MM(
                    Mult(MM($c_{t-1}$),MM($x_t$))
                ),
                MM($x_t$)
            ),
            Sigmoid(
                Add( MM($x_t$), MM($h_{t-1}$) )
            )
        )
    ),
    $h_{t-1}$,
    Sigmoid(
        Add( MM($x_t$), MM($h_{t-1}$) )
    )
)|12
```

### 1. ARCHITECTURE OPTIMIZATIONS DURING DSL COMPILATION

To improve the running speed of the RNN cell architectures, we can collect all matrix multiplications performed on a single source node ($x_t$, $x_{t-1}$, $h_{t-1}$, or $c_{t-1}$) and batch them into a single matrix multiplication. As an example, this optimization would simplify the LSTM's 8 small matrix multiplications (4 for $x_t$, 4 for $h_{t-1}$) into 2 large matrix multiplications. This allows for higher GPU utilization and lower CUDA kernel launch overhead.

### 2. PREVENTING DUPLICATES THROUGH CANONICAL ARCHITECTURE ORDERING

There exist many possible DSL specifications that result in an equivalent RNN cell. When two matrix multiplications are applied to the same source node, such as $Add(MM(x_t), MM(x_t))$, a matrix multiplication reaching an equivalent result can be achieved by constructing a specific matrix and calculating $MM(x_t)$. Additional equivalences can be found when an operator is commutative, such as $Add(x_t, h_{t-1})$ being equivalent to the reordered $Add(h_{t-1}, x_t)$. We can define a canonical ordering of an architecture by sorting the arguments of any commutative nodes. In our work, nodes are sorted according to their DSL represented as a string, though any consistent ordering is allowable. For our core DSL, the only non-commutative operation is $Gate3$, where the first two arguments can be sorted, but the input to the gate must remain in its original position. For our extended DSL, the $Sub$ and $Div$ operators are order sensitive and disallow any reordering.

## APPENDIX B: LANGUAGE MODELING

### 1. BASELINE EXPERIMENTAL SETUP AND HYPERPARAMETERS

The models are trained using stochastic gradient descent (SGD) with an initial learning rate of 20. Training continues for 40 epochs with the learning rate being divided by 4 if the validation perplexity has not improved since the last epoch. Dropout is applied to the word embeddings and outputs of layers as in Zaremba et al. (2014) at a rate of 0.2. Weights for the word vectors and the $softmax$ were also tied (Inan et al., 2016; Press and Wolf, 2016). Aggressive gradient clipping (0.075) was performed to ensure that architectures such as the ReLU RNN would be able to train without exploding gradients. The embeddings were initialized randomly between $[-0.04, 0.04]$.

During training, any candidate architectures that experienced exploding gradients or had perplexity over 500 after five epochs were regarded as failed architectures. Failed architectures were immediately terminated. While not desirable, failed architectures still serve as useful training examples for the ranking function.

### 2. RANKING FUNCTION HYPERPARAMETERS

For the ranking function, we use a hidden size of 128 for the TreeLSTM nodes and a batch size of 16. We use $L2$ regularization of $1 \times 10^{-4}$ and dropout on the final dense layer output of 0.2. As we are more interested in reducing the perplexity error for better architectures, we sample architectures more frequently if their perplexity is lower.

For unrolling of the architectures, a proper unroll would replace $x_t$ with $x_{t-1}$ and $x_{t-1}$ with $x_{t-2}$. We found the ranking network performed better without these substitutions however and thus only substituted $h_{t-1}$ to $h_{t-2}$ and $c_{t-1}$ to $c_{t-2}$.

### 3. HOW BASELINE EXPERIMENTAL SETTINGS MAY IMPAIR ARCHITECTURE SEARCH

The baseline experiments that are used during the architecture search are important in dictating what models are eventually generated. As an example, BC3 may not have been discovered if we had used all the standard regularization techniques in the baseline language modeling experiment. Analyzing how variational dropout (Gal and Ghahramani, 2016) would work when applied to BC3 frames the importance of hyperparameter selection for the baseline experiment.

On LSTM cells, variational dropout (Gal and Ghahramani, 2016) is only performed upon $h_{t-1}$, not $c_{t-1}$, as otherwise the long term hidden state $c_t$ would be destroyed. For BC3, equation 6 shows that the final gating operation mixes $c_t$ and $h_{t-1}$. If variational dropout is applied to $h_{t-1}$ in this equation, BC3's hidden state $h_t$ will have permanently lost information. Applying variational dropout only to the $h_{t-1}$ values in the two gates $f$ and $o$ ensures no information is lost. This observation provides good justification for not performing variational dropout in the baseline experiment given that this architecture (and any architecture which uses $h_{t-1}$ in a direct manner like this) would be disadvantaged otherwise.

### 4. HYPERPARAMETERS FOR PTB AND WIKITEXT-2 BC3 EXPERIMENTS

For the Penn Treebank BC3 language modeling results, the majority of hyper parameters were left equal to that of the baseline AWD-LSTM. The model was trained for 200 epochs using NT-ASGD with a learning rate of 15, a batch size of 20 and BPTT of 70. The variational dropout for the input, RNN hidden layers, and output were set to 0.4, 0.25, and 0.4 respectively. Embedding dropout of 0.1 was used. The word vectors had dimensionality of 400 and the hidden layers had dimensionality of 1080. The BC3 used 3 layers with weight drop of 0.5 applied to the recurrent weight matrices. Activation regularization and temporal activation regularization of 2 and 2 were used. Gradient clipping was set to 0.25. Finetuning was run for an additional 13 epochs. For the WikiText-2 BC3 language modeling results, the parameters were kept equal to that of the Penn Treebank experiment. The model was run for a total of 100 epochs with 7 epochs of finetuning.

For the Penn Treebank GRU language modeling results, the hyper parameters were equal to that of the BC3 PTB experiment but with a hidden size of 1350, weight drop of 0.3, learning rate of 20, and gradient clipping of 0.15, and temporal activation regularization of 1. The model was run for 280 epochs with 6 epochs of finetuning. For the WikiText-2 GRU language modeling results, the hyper parameters were kept equal to those of the Penn Treebank experiment. The model was run for 125 epochs with 50 epochs of finetuning where the weight drop was reduced to 0.15.

## APPENDIX C: MACHINE TRANSLATION

### 1. BASELINE EXPERIMENTAL SETUP AND HYPERPARAMETERS FOR RL

To represent an architecture with the encoder, we traverse through the architecture recursively, starting from the root node. For each node, the operation is tokenized and embedded into a vector. An LSTM is then applied to this vector as well as the result vectors of all of the current node's children. Note that we use the same LSTM for every node but reset its hidden states between nodes so that it always starts from scratch for every child node.

Based on the encoder's vector representation of an architecture, the action scores are determined as follows:

$$\text{action scores} = softmax(linear(LSTM(ReLU(linear(\text{architecture encoding})))))$$

We then choose the specific action with a multinomial applied to the action scores. We encourage exploration by randomly choosing the next action according to an epsilon-greedy strategy with $\epsilon = 0.05$.

The reward that expresses how well an architecture performed is computed based on the validation loss. We re-scale it according to a soft exponential so that the last few increases (distinguishing a good architecture from a great one) are rewarded more. The specific reward function we use is $R(loss) = 0.2 \times (140 - loss) + 4^{0.3815 \times (140 - loss) - 50}$ which follows earlier efforts to keep the reward between zero and 140.

For pre-training the generator, our list of priors are:

1. to maintain a depth between 3 and 11
2. use the current input $x_t$, the hidden state $h_{t-1}$, at least one matrix multiplication (MM) and at least one activation function
3. do not use the same operation as a child
4. do not use an activation as a child of another activation
5. do not use the same inputs to a gate
6. that the matrix multiplication operator (MM) should be applied to a source node

### 2. FULL DSL FOR DEEPLY NESTED $Gate3$ AND RESIDUAL $x_t$ WITH POSITIONAL ENCODING

For obvious reasons these DSL definitions would not fit within the result table. They do give an indication of the flexibility of the DSL however, ranging from minimal to quite complex!

**5-deep nested** $Gate3$ **with *LayerNorm***

```
Gate3(Gate3(Gate3(Gate3(Gate3(LayerNorm(MM(Tanh(MM(Var('hm1')))))), MM(
    Tanh(MM(Tanh(Tanh(MM(Var('hm1'))))))), Sigmoid(MM(Var('hm1')))), Var
    ('hm1'), Sigmoid(MM(Var('hm1')))), Var('hm1'), Sigmoid(MM(Var('hm1'))
    )), Var('hm1'), Sigmoid(MM(Var('hm1')))), Var('x'), Sigmoid(MM(Var('
    hm1'))))
```

**Residual** $x_t$ **with positional encoding**

```
Gate3(Gate3(Var('fixed_posenc'), Var('hm1'), Sigmoid(MM(Tanh(MM(Tanh(MM(
    Tanh(MM(Tanh(Var('hm1')))))))))))), Var('x'), Sigmoid(MM(Tanh(MM(Tanh(
    MM(Var('hm1')))))))))
```

### 3. HYPER PARAMETERS FOR THE IWSLT'16 MODELS

For the models trained on the IWSLT'16 English to German dataset, the hyper parameters were kept largely equivalent to that of the default OpenMT LSTM baseline. All models were unidirectional and the dimensionality of both the word vectors and hidden states were 600, required as many of the generated architectures were residual in nature. The models were 2 layers deep, utilized a batch size of 64, and standard dropout of 0.2 between layers. The learning rate began at 1 and decayed by 0.5 whenever validation perplexity failed to improve. When the learning rate fell below 0.03 training was finished. Models were evaluated using a batch size of 1 to ensure RNN padding did not impact the results.

### 4. FINDING AN ARCHITECTURE FOR THE DECODER AND ENCODER/DECODER

We also briefly explored automated architecture generation for the decoder as well as for the encoder and decoder jointly which yielded good results with interesting architectures but performances fell short of the more promising approach of focusing on the encoder.

With additional evaluated models, we believe both of these would yield comparable or greater results.

## 5. VARIATION IN ACTIVATION PATTERNS BY GENERATED ARCHITECTURES

Given the differences in generated architectures, and the usage of components likely to impact the long term hidden state of the RNN models, we began to explore the progression of the hidden state over time. Each of the activations differs substantially from those of the other architectures even though they are parsing the same input.

As the input features are likely to not only be captured in different ways but also stored and processed differently, this suggests that ensembles of these highly heterogeneous architectures may be effective.

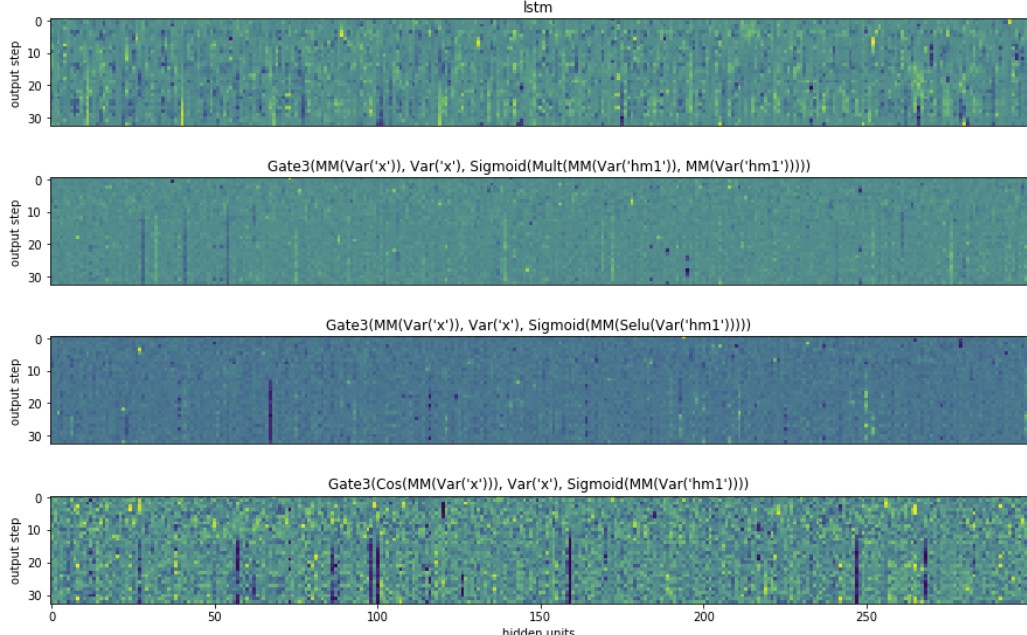

Figure 6: Visualization of the hidden state over time for a variety of different generated architectures.

## 6. EXPLORATION VERSUS EXPLOITATION

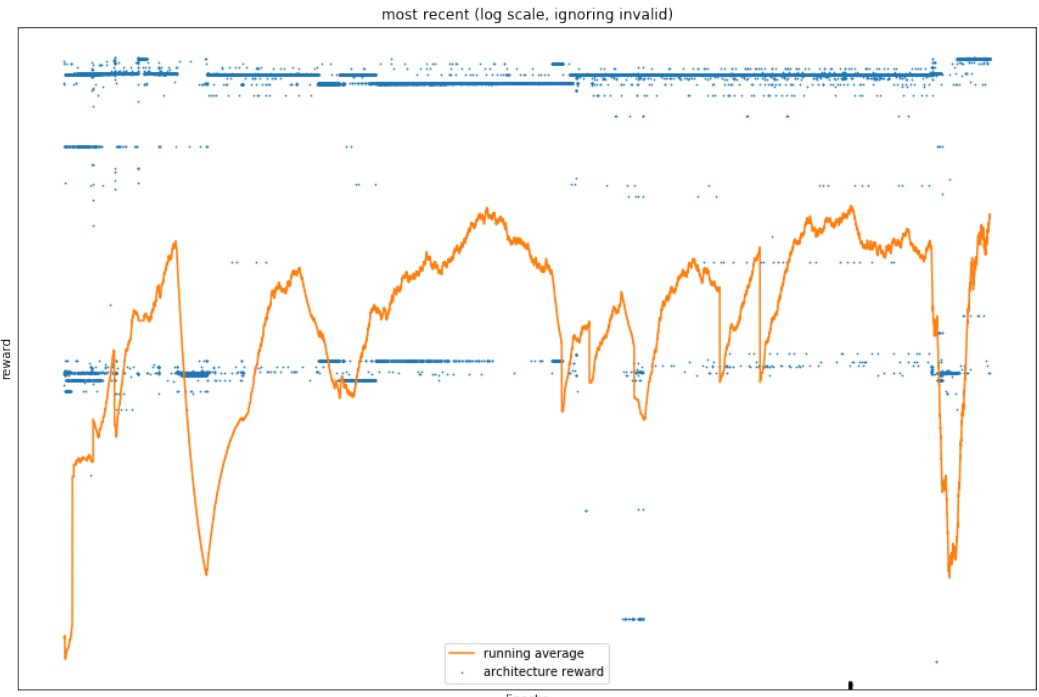

Figure 7: This figure shows the progress of the generator over time, highlighting the switches between exploitation (increasing running average up to plateau) and exploitation (trying out new strategies and thus decreasing running average at first). Only valid architectures are shown. Higher reward is better with the x-axis showing progress in time.

## 7. INFRASTRUCTURE DETAILS

We ran the architecture search for 5 days on one CPU head node and several worker nodes with a total of 28 GPUs on AWS. 24 GPUs were of the type Tesla K80 and 4 GPUs were of the type Tesla M40. The best architecture (Table 3, bottom row) was found after 40 hours. However, as evident in Figure 6, the generator created well-performing architectures more and more consistently with more training.

