# OpenReview forum: "A Flexible Approach to Automated RNN Architecture Generation"
_ICLR.cc/2018/Conference — Invite to Workshop Track_

### Official Review · AnonReviewer2 · 2017-11-10
**Nice work, but limited in scope and does not appear to generalize well**

**Rating:** 4
**Confidence:** 4

**Review:**

This work tries to cast the search of good RNN Cell architectures as a black-box optimization problem where examples are represented as an operator tree and are either 1. Sampled randomly and scored based on a learnt function OR 2. Generated by a RL agent.
While the overall approach appears to generalize previous work, I see a few serious flaws in this work:

Limited scope
As far as I can tell this work only tries to come up with a design for a single RNN cell, and then claims that the optimality of the design will carry over to stacking of such modules, not to mention more complicated network designs.
Even the design of a single cell is heavily biased by human intuition (section 4.1) It would have been more convincing to see that the system learn heuristics such as “don’t stack two matrix mults” rather than have these hard coded.

No generalization guarantees:
No attempt is made to optimize hyperparameters of the candidate architectures. This leads one to wonder if the winning architecture has won only because of the specific parameters that were used in the evaluation.
Indeed, the experiments in the paper show that a cell which was successful on one task isn’t necessary successful on a different one, which questions the competence of the scoring function / RL agent.

On the experimental side:
No comparison is made between the two optimization strategies, which leaves the reader wondering which one is better.
Control for number of network variables is missing when comparing candidate architectures.

---

> ### Author Response · Authors · 2018-01-05
> **Thank you for your review**
>
> Thanks for your response. While we agree that our work helps generalize previous work in this area we do also agree that it doesn’t resolve all the issues that you’ve note.
>
> In terms of limited scope, our work does show that a single RNN cell trained on a two layer setup is able to extend to a three layer setup, as seen with our best BC3 cell results being reported on a three layer setup. Our baseline experimental setup involved a two layer cell setup for this very reason, to ensure that discovered RNN cells could be stacked. We believe we show this is indeed true.
>
> The heuristics introduced for that section were introduced to limit the search space. Whilst it is likely that the architecture generator would learn to avoid models which our heuristics filtered we decided that the computation expended for learning those relatively simple concepts was better spent on the architecture search process itself.
>
> Whilst we kept the number of parameters equal for the language modeling experiment, keeping the number of parameters equal for the translation experiment was more complicated. The RNN cells discovered residual connections and hence prevented easy scaling up or down of the overall hidden size without an additional projection layer.
>
> In terms of generalization guarantees, like other machine learning models the architecture generator is only informed by the training data it receives. In our instance this is from a single task with limited sized models due to the computational constraints but this could be extended to larger models and across more varied tasks. We agree that the competence of the scoring function / RL agent would be dependent on the training data it receives.

---

> > ### Comment · AnonReviewer2 · 2018-01-09
> > **Thanks for the reply**
> >
> > This work is indeed nice, but I still think it is not very useful for ML practitioners, for the reasons I mentioned: limited scope, no generalization across hyper-parameters (which I believe we agree on).
> > I also don't think this method contributes to theoretical understanding of RNNs.

---

### Official Review · AnonReviewer1 · 2017-11-27

**Rating:** 5
**Confidence:** 4

**Review:**

This paper investigates meta-learning strategy for automated architecture search in the context of RNN. To constraint the architecture search space, authors propose a DSL that specifies the RNN recurrent operations. This DSL allows to explore RNN architectures using either random search or a reinforcement-learning strategy. Candidate architectures are ranked using a TreeLSTM that tries to predict the architecture performances. The top-k architectures are then evaluated by fully training them on a given task.

Authors evaluate their approach on  PTB/Wikitext 2 language modeling and Multi30k/IWSLT'16  machine translation. In both experiments, authors show that their approach obtains competitive results and can sometime outperforms RNN cells such as GRU/LSTM. In the PTB experiment, their architecture however underperforms other LSTM variant in the literatures.


- Quality/Clarity
The paper is overall well written and pleasant to read.

Few details can be clarified. In particular how did you initialize the weight and bias for both the LSTM/GRU baselines and the found architectures? Is there other works leveraging RNN that report results on the Multi30k/IWSLT datasets?

You state in paragraph 3.2 that human experts can inject the previous best known architecture when training the ranking networks. Did you use this in the experiments? If yes, what was the impact of this online learning strategy on the final results?


- Originality
The idea of using DSL + ranking for architecture search seems novel.


- Significance
Automated architecture search is a promising way to design new networks. However, it is not clear why the proposed approach is not able to outperforms other LSTM-based architectures on the PTB task. Could the problem arise from the DSL that constraint too much the search space ? It would be nice to have other tasks that are commonly used as benchmark for RNN to see where this approach stand.

In addition, authors propose both a DSL, a random and RL generator and a ranking function. It would be nice to disentangle the contributions of the different components. In particular, did the authors compare the random search vs the RL based generator or the performances of the RL-based generator when the ranking network is not used?

Although authors do show that they outperform NAScell in one setting, it would be nice to have an extended evaluation (using character level PTB for instance).

---

> ### Author Response · Authors · 2018-01-05
> **Thank you for your review**
>
> Thank you for your review.
>
> In regards to initialization of the weights and bias within our models, we used the default that was set within PyTorch. For both weights and bias this was uniform sampling from +-(1/sqrt(HIDDEN)).
>
> We are not aware of results that utilize only an RNN on the Multi30k/IWSLT datasets, likely as the LSTM or GRU are considered the standard baselines for such work, consistently outperforming RNNs.
>
> We did not inject human known architectures into our search as we were interested in understanding whether the given architecture search process could generate architectures of comparable accuracy. Extending existing architectures would be an interesting extension and could provide an even more computationally efficient starting point for the ranking function’s search.
>
> For the LM task, the LSTM has been finely tuned over quite some time for these tasks. The work of Melis et al and Merity et al go into quite an amount of detail about the hyperparameter search they perform, with the former leveraging large scale automated hyperparamter optimization on Google’s infrastructure and the latter featuring substantial manual tuning building on the work of many others.
>
> Another important consideration that we believe may have been problematic for beating the LSTM’s performance is that the baseline experiments we utilized featured smaller models for faster training time. Whilst necessary for our setup it is possible that larger models with fewer iterations would have been a better choice.

---

### Official Review · AnonReviewer3 · 2017-11-27
**The authors present an interesting framework to search for new RNN models, with some promising results.**

**Rating:** 6
**Confidence:** 4

**Review:**

The authors introduce a new method to generate RNNs architectures. The authors propose a domain-specific language two types of generators (random and RL-based) together with a ranking function and evaluator. The results are promising, and this research introduces a framework that might enable the community to find new interesting models. However, it's not clear how these framework compare with previous ones (see below). Also, the clarity of the text could be improved.

Pros:
1. An interesting automatic method for generating RNNs is introduced by the authors (but is not entirely clear how does ir compare with previous different approaches)
2. The approach is tested in a number of tasks: Language modelling (PTB and wikipedia-2) and machine translation (
3. In these work the authors tested a wide range of different RNNs
3. It is interesting that some of the best performing architectures (e.g. LSTM, residual nets) are found during the automatic search


Cons:
1. It would be nice if the method didn’t rely on defining a specific set of functions and operators upon which the proposed method works.
2. The text has some typos: for example: “that were used to optimize the generator each batch”
3. In section 5, the authors briefly discuss other techniques using RL and neuroevolution, but they never contrast these approaches with theirs. Overall, it would be nice if the authors had made a more direct comparison with other methods for generating RNNs.
4. The description of the ranking function is not clear. What kind of networks were used? This appears to introduce a ranking-network-specific bias in the search process.

Minor comments:
1. The authors study the use of subtractive operators. Recently a new model has considered the use of subtractive gates in LSTMs as a more biologically plausible implementation (Cortical microcircuits as gated-recurrent neural networks, NIPS 2017).
2. Figure 4 missing label on x-axis
3. End of Related Work period is missing.
3. The authors state that some of the networks generated do not follow human intuition, but this doesn’t appear to discussed. What exactly do the authors mean?
4. Not clear what happens in Figure 4 in epoch 19k or so, why such an abrupt change?
5. Initial conditions are key for such systems, could the init itself be included in this framework?

---

> ### Author Response · Authors · 2018-01-05
> **Thank you for your review.**
>
> Thank you for your review.
>
> We agree that a more flexible set of operators is highly desirable, especially for finding radically novel architectures. Here, we tried to find a trade-off between flexible operators and a reasonably sized search space. In the future, expanding that search space even further will be an interesting avenue to find even more diverse architectures.
>
> The recent work regarding biologically plausible models that utilize subtractive gates is fascinating - thanks for the reference!
>
> For the ranking function, all the nodes except for those that are positional dependent are ChildSum TreeLSTM nodes, whilst those nodes requiring positional information are N-ary TreeLSTM nodes (more detail in 3.2). The ranking function hyper parameters and details are described in greater detail in Appendix B2.
>
>
> In regards to Figure 4, epoch 19k, we are not entirely certain what occurred there. This was a continuous run and there were no explicit changes during that section. In the text, we briefly mention the hypothesis that the generator first learns to build robust architectures and is only then capable of inserting more varied operators without compromising the RNN's overall stability.
>
> For the initialization, it is the default that was found within PyTorch. As an example, the initializations for the RNNs were all equivalent to PyTorch’s Linear, which performs uniform initialization between +-(1/sqrt(HIDDEN)).
> https://github.com/pytorch/pytorch/blob/b06276994056ccde50a6550f2c9a49ab9458df8f/torch/nn/modules/linear.py#L48

---

> > ### Comment · AnonReviewer3 · 2018-01-12
> > **still not clear what's the advantage over existing methods**
> >
> > Thanks for your reply and clarifications.
> >
> > One of the key issues that the authors did not address in their reply is that of the comparison with previous work, this is of importance to properly assess the potential impact of this work. Therefore, I have decided to revise my rating slightly down. It is therefore, unlikely that this is going to be accepted. However, I encourage the authors in finishing this work with our comments in mind.

---

> > > ### Author Response · Authors · 2018-01-15
> > > **Advantages are outlined in Section 5: Related Work**
> > >
> > > From your previous comment, it was not clear to us what parts you specifically wanted to have contrasted with previous work - we do have a section that highlights what we think are our major improvements and distinctions over previous work. If you could please clarify what previous work you want us to discuss in more detail, that would be greatly appreciated and we are happy to extend the paper in that direction.
> > > To reiterate, we see Zoph and Le (2017) as the most similar approach to ours and improve over that with two major contributions: 1. we introduce a domain-specific-language (DSL) that allows for the generation of much more flexible and thus more radically novel architectures (see Section 5, paragraph 1 for details) and 2. we expand on commonly used operators with so far widely unexplored operators like sine curves, division and others.

---

> > > > ### Comment · AnonReviewer3 · 2018-01-15
> > > > **quantitative comparison**
> > > >
> > > > Thanks for clarifying this point. I understand that the manuscript contains a section briefly highlighting the diferences, but to properly evaluate what does the community gain with this method. Therefore having a more quantitative comparison would be important.

---

> > > > > ### Author Response · Authors · 2018-01-16
> > > > > **hard to make quantitative comparisons other than results**
> > > > >
> > > > > Did you have a specific quantitative comparison in mind? We do compare with the architectures found by e.g. Zoph and Le (2017) in Table 1.
> > > > > We agree that it would be great to have a metric "radical novelty of architecture" or similar but it seems like at this point the qualitative differences of architectures with novel operators and the flexibility of our DSL along with good quantitative results are the key metrics that we can report.
> > > > > We also updated the paper to report our hardware infrastructure and the specific computation times to make our work better comparable to existing methods.

---

> > > > > > ### Comment · AnonReviewer3 · 2018-01-22
> > > > > > **comparing the different algorithms**
> > > > > >
> > > > > > Agree that is not straight forward, but would be in my view important so that the community understands what do we gain with a new method. One could for example run the different methods (for different initial conditions) and quantify how often they end up in better solutions (or 'radically' new solutions).

---

### Decision · Program_Chairs · 2018-01-29
**ICLR 2018 Conference Acceptance Decision**

**Decision:**

Invite to Workshop Track

**Comment:**

The paper presents a domain-specific language for RNN architecture search, which can be used in combination with learned ranking function or RL-based search. While the approach is interesting and novel, the paper would benefit from an improved evaluation, as pointed out by reviewers. For example, the paper currently evaluated coreDSL+ranking for language modelling and extendedDSL+RL for machine translation. The authors should use the same evaluation protocol on all tasks, and also compare with the state-of-the-art MT approaches.